# Changes in Antioxidant and Color Properties of Raisins According to Variety and Drying Method

**Mehmet Settar Ünal [1],\*, Emrah Güler [2]** and **Mehmet Yaman [3]**

1   Department of Horticulture, Faculty of Agriculture, Sirnak University, Sirnak 73000, Turkey
2   Department of Horticulture, Faculty of Agriculture, Bolu Abant Izzet Baysal University, Bolu 14030, Turkey; emrahguler6@gmail.com
3   Department of Horticulture, Faculty of Agriculture, Erciyes University, Kayseri 38030, Turkey; mehmetyaman@erciyes.edu.tr
\*   Correspondence: munal62@hotmail.com

**Abstract:** Raisins are a significant grape product with an annual trade volume of approximately USD 2 billion. There are various methods employed for drying grapes into raisins. This study aimed to investigate the effects of different drying methods on raisins, specifically, without pretreatment (SD), treatment with potassium carbonate (5%) containing 2% olive oil (POT), and treatment with ashy water with 2.5% olive oil (AOW). The study examined the changes in raisin weight, physicochemical characteristics, total phenolics, antioxidant and anthocyanin contents, color properties, and protein content in three grape varieties: Deyvani, Haseni, and Reşek. The results indicated that POT treatment resulted in the lightest raisins in Deyvani (0.48 g) and Reşek (0.58 g), while Haseni exhibited the heaviest raisins (0.64 g) under the same treatment. The variations in physicochemical characteristics were relatively limited compared to the changes observed in phenolic and antioxidant properties. Nontreated raisins had the lowest anthocyanin content across all varieties. Protein content was significantly affected by the POT treatment, while the effects of the AOW and SD treatments were comparatively minor. SD drying resulted in slightly brighter raisins, but lower phenolic content and significantly lower color properties. The findings of this study highlight the differential effects of drying methods on different grape varieties, emphasizing the importance of considering a variety-specific approach when selecting the drying method.

**Keywords:** grape; potassium carbonate; oak ash; total phenolic content; color changes; DPPH scavenging activity; multivariate approaches

## 1. Introduction

Grapes hold a prominent position among the crops cultivated worldwide, boasting a substantial production volume. Furthermore, processed grape products play a significant role in agricultural trade [1]. Notably, wine stands out as the leading exported commodity, with a quantity of 10.6 million tons, followed by fresh grapes at 4.8 million tons. The import value aligns closely with the export value. In terms of the sector's trade, the wine market amounts to approximately USD 39 billion, while fresh grapes contribute around USD 10 billion. Raisins, with a trade value of roughly USD 2 billion, exhibit a similar balance between exports and imports [2].

The process of obtaining raisins typically involves pre-drying, drying, and post-drying procedures, which significantly impact the quality of the raisins in terms of enzyme activities, sugar content, and drying time [3,4]. The drying process leads to substantial changes in total antioxidant activity, volatile compounds, vitamins, minerals, and fiber content compared to fresh grapes [5]. Different methods can be employed to dry fresh grapes, including sunlight/shade drying, immersion in potash or water with ash, or modern techniques such as vacuum or microwave drying and hot air drying [6]. In Turkey, grapes are dried using various methods based on consumer preferences. Sun drying

involves placing the grapes on soil, concrete floors, or shelves and allowing them to dry for 2–3 weeks under direct sunlight. Shade drying, on the other hand, involves drying the grapes for 2 or 4 weeks without exposure to direct sunlight. Prior to drying, pretreatment can be applied to remove water from the grapes [7]. In Asia Minor, a solution is used to enhance the drying rate of grape clusters. In ancient times, solutions were prepared using wood ash or olive oil, while nowadays, olive oil, wood ash, or potassium carbonate are commonly used. Commercial cold dips often include a combination of potassium carbonate and fatty acids known as potash solution. Water containing wood ash is also used as an organic dipping solution. Grapes dried without dipping exhibit a blackish-gray color, tough skin, a dry and oil-free surface, and lower sugar content compared to those dried with dipping methods [8].

The consumption of grapes and raisins can be traced back to ancient times, even predating recorded history [9]. It is believed that early hunter-gatherer societies recognized the desirable qualities of wild grapes and observed that grapes could naturally transform into a dried, edible form when they fell off the vine and were exposed to sunlight [10]. The utilization of raisins as both food and decorative elements can be found in ancient murals of the Mediterranean region and archaeological discoveries from the Bronze Age, such as those found at Lachish in Israel. Raisins gained recognition and value due to their ease of storage and transport [11]. Grapes are typically categorized as juicy berries. Their high juice content makes them highly perishable during the postharvest period due to the combination of abundant moisture and sugar, which leads to a shorter shelf life [12]. Consequently, once harvested, fresh grapes become susceptible to physical damage and a decline in quality. When stored under normal conditions, the deterioration of fresh grapes occurs rapidly [13]. Hence, it is crucial to either consume them promptly or process them into various products to minimize postharvest losses in terms of quantity and quality [14].

In recent years, people's demand for healthy food products has been increasing rapidly [15–17]. In the past, consumers primarily considered taste, cost, and availability when deciding to purchase fresh or processed agricultural products. However, the growing body of research highlighting the impact of diet on human health and nutrition has led to an increasing number of consumers making their purchasing decisions based on the nutritional value and potential health benefits of food [18,19]. Plant scientists and breeders work intensively to identify and breed species and varieties rich in biochemical substances that positively contribute to health. Numerous studies have been published recently examining the quality and biochemical contents of grapes [20–22]. Similarly, it is possible to come across studies in the literature on the content of raisins [23,24]. Raisins are an important source of micro- and macronutrients, providing sugars, vitamins, minerals, and fiber. Additionally, they contain a diverse array of bioactive compounds such as polyphenols [25]. Raisins have a sugar content of over 62%, primarily composed of monosaccharides, glucose, and fructose, which are present in nearly equal proportions. The amount of sucrose in raisins is comparatively low, providing easily absorbed energy and a moderate glycemic index [26]. However, almost 95% of the world's raisin production is supplied with Sultani Çekirdeksiz, the synonym of Thompson Seedless [27]. Although this variety dominates the market, Breska et al. reported that Sultani Çekirdeksiz and Fiesta, the other widely produced variety, are the lowest among 16 white grape varieties in terms of phenolic content and antioxidant activity [28]. In addition, Sultani Çekirdeksiz is not a variety that can be grown in every geoclimate. Determining the dried grape varieties suitable for the current geoclimatic conditions is a crucial issue [29]. Since species and varieties have different responses to processing methods and technologies [30], studying suitable drying methods for specific varieties is vital for obtaining raisins of the desired quality.

This study investigated the effects of different drying methods on the physicochemical, antioxidant, and colorimetric properties of seeded grape varieties adapted to high altitudes in Turkey.

## 2. Materials and Methods

### 2.1. Plant Material

This study was carried out on Deyvani, Haseni, and Reşek grape (*Vitis vinifera* L.) varieties grown in a producer's vineyard in the Midyat district of Mardin Province, Türkiye (37°25′30.2″ N 41°22′45.8″ E, and 950 m elevation). The varieties were hand-harvested when the total soluble solid (TSS) content reached 22%, in mid-August for Haseni and Reşek and at the end of August for Deyvani. Clusters with uniform berry sizes were chosen for the study.

### 2.2. Pretreatment Solution Preparation and Treatment

An adequate number of grapes was collected from each variety at the beginning. The grapes were then divided into three groups for each pretreatment solution. The preparation of the pretreatment solutions was as follows:

Preparation of water with ash: water comprising 7% oak wood ash (*w/v*) was brought to a boil. It was then removed from heat and left to cool, allowing the ash to settle at the bottom. The water was separated and boiled again with approximately 2.5% olive oil added.

Preparation of potassium carbonate: a solution was created by dissolving 5 kg of potassium carbonate in 100 L of water, and 1.5 L of high-acidity olive oil (2%) was added [31]. Subsequently, the grape clusters were immersed in the prepared solution for approximately 5 s. All samples were then placed on a white cloth spread on soil to prevent contamination and were dried in the sun. Regular checks were conducted during the drying process to ensure uniform drying. Drying continued until the moisture content of the samples reached 17% [32]. Throughout the drying process, the average temperature in the region was 31.6 °C, with a relative air humidity of 16%, no precipitation, and a wind speed of 0.6 m/s.

The control group (sun-dried) involved samples without any pretreatment solution, while the immersion of grape clusters in potassium carbonate and water with ash solutions was referred to as POT and AOW applications, respectively.

### 2.3. Measurement of Raisin Weight and Physicochemicals

Raisin weight was assessed by measuring 50 raisins in each replication using digital scales. For physicochemical analyses, 10 g samples of raisins were taken from each replication, their seeds were removed, and the samples were blended using an electric blender and combined with 100 mL of distilled water. They were then left at room temperature for 6 h. Subsequently, the mixture was filtered using cheesecloth, and an appropriate amount of the resulting filtrate was collected. The TSS was determined using a digital refractometer (ATC, 0–32, İstanbul, Turkey). To determine titratable acidity, an appropriate volume of the filtrate was titrated with 0.1 N NaOH [32]. The pH of the filtrate was measured using a pH meter (Orion Star A211, Thermo Scientific, Waltham, MA, USA).

### 2.4. Total Phenolics and DPPH Scavenging Assay

The microscale procedure reported by Waterhouse [33] was used with some modifications in the determination of the total phenolic content. Briefly, 1600 μL of distilled water and 50 μL of Folin–Ciocalteu agent were added to 50 μL of methanolic extract and mixed gently. Then, 300 μL of 7% (*w/v*) calcium carbonate solution was added and vortexed. After the mixture was left in the dark under room conditions for 2 h, its absorbance at 760 nm was read using a UV-Vis spectrophotometer (SP-UV1100, DLAB, Beijing, China). Obtained absorbance values were converted to real content by calculating the equation obtained with a standard curve ($R^2 = 0.99$) prepared using 0.5, 1, 2, 3, 4, 5, and 6 mM gallic acid with the same procedure.

An ethanol solution of 2,2 Diphenyl 1 picrylhydrazyl obtained from Sigma-Aldrich (Schnelldorf, Germany) was prepared with a final absorbance within the range of 0.7–0.8 to measure the DPPH scavenging activity. The activity was measured by determining the most

appropriate methanolic extract amount by preliminary trials, with a final volume of 2 mL; a 50 μL sample, 1450 μL of ethanol, and 500 μL of DPPH solution were added sequentially and vortexed. The prepared solution was measured at the 520 nm wavelength in a UV-Vis spectrophotometer after 15 min, and the DPPH scavenging capacity was calculated with the following formula.

$$DPPH\ (\%) = (A_{blank} - As_{ample})/A_{blank}.$$

### 2.5. Soluble Protein and Total Monomeric Anthocyanin Assays

The total amount of soluble protein was determined using the modified Bradford method [34]. Briefly, 1950 μL of Bradford agent was added to 50 μL of methanolic extract. The absorbance was read at the 595 nm wavelength on a UV-Vis spectrophotometer, and the actual content was calculated with the equation obtained from the standard curve. Solutions obtained by serial dilution of 100 mg/mL bovine serum albumin stock solution were used in preparing the standard curve.

For the determination of total monomeric anthocyanin, the pH differential method defined by Fuleki and Francis [35] was applied with some modifications. Briefly, the appropriate amount to be taken from the methanolic extract was determined and the dilution factor was recorded in order to obtain an absorbance in the range of 0.4–0.8 at the maximum wavelength. Subsequently, 0.4 mL of prepared dilution was placed into 2 separate tubes and filled to 2 mL with pH 1.0 and pH 4.5 buffer solutions. The tubes were then capped and kept in the refrigerator for 2 h in dark conditions. Samples were measured at the wavelengths of 516 nm and 700 nm, and the true absorbance was calculated with the following formula:

$$Absorbance\ (A) = (A_{516} - A_{700})\ pH\ 1. - (A_{516} - A_{700})\ pH\ 4.5.$$

The total amount of monomeric anthocyanin was calculated by adding the obtained absorbance value to the formula below.

$$Total\ anthocyanin\ (mg/L) = (A \times 10^3 \times MW \times DF)/(E \times L)$$

A: absorbance, MW: molecular weight of pigments, (cyanidin 3 glucoside; 484.83 g/mol), DF: dilution factor, E: molar absorbance (26,900), L: optical path (1 cm) of the cuvette.

### 2.6. Colorimetric Properties and Color Density Determination

Numerical color values in CIE color space L*, a*, b*, Chroma and Hue angle values were determined with a handheld colorimeter (PCE CSM-4, Southampton, UK). The anthocyanin degradation method was used to determine the color intensity and polymeric color [35]. In sample preparation, dilution, and absorbance values were adjusted as in total monomeric anthocyanin, and 0.4 mL samples were placed in two different tubes, one filled with distilled water and the other with 20% (*w/v*) metabisulfite solution. Both solutions were measured half an hour later with a UV-Vis spectrophotometer at 420 nm, 516 nm, and 700 nm. The color density was calculated in the untreated sample, and the polymeric color was computed in the bisulfite-treated sample with the following formula.

$$Color\ Density/Polymeric\ color = [(A_{516} - A_{700}) + (A_{420} - A_{700})]\ (DF).$$

DF: dilution factor

### 2.7. Statistical Evaluations

The study was carried out in triplicate for each treatment and variety in a factorial trial design. Data were subjected to Levene's homogeneity test to assure that they were homogeneous. After determining the normality, data were subjected to a two-way analysis of variance (ANOVA) to determine the effects of the factors and their interaction. The

differences between the means of the factors and interactions were compared using a Student's *t*-test (LSD) with an alpha level of 0.05. The interrelations among the drying methods, varieties, and studied traits were evaluated by principal component (PCA) and heatmap analyses using the "ggplot2" package of R Studio [36].

## 3. Results

### 3.1. Raisin Weight and Physicochemicals

The weights of raisins showed significant variation based on different factors, including variety ($p < 0.001$), drying method ($p < 0.05$), and their interactions ($p < 0.001$). The Reşek variety yielded the heaviest raisins, weighing 0.662 g, whereas the Deyvani variety produced the lightest ones, weighing 0.541 g. Among the drying methods, SD did not show a significant difference compared to the others, while AOW (0.618 g) had a notably higher mean raisin weight than POT (0.569 g). The highest mean weight of the berries was observed in the Reşek variety with AOW treatment, measuring 0.779 g. Considering the interactive effect, POT resulted in the lowest mean weight values for the Reşek (0.579 g) and Deyvani (0.483 g) varieties, while it yielded the highest mean weight in the Haseni (0.646 g) variety. SD yielded the highest raisin weight in Deyvani (0.577 g), while AOW-treated Reşek obtained the highest raisin weights with a mean of 0.779 g (Table 1).

**Table 1.** Changes in raisin weight and physicochemical characteristics according to variety and drying method (mean ± standard deviation).

| Variety | | Raisin Weight (g) | TSS (%) | pH | TA (mg/L) |
|---|---|---|---|---|---|
| | Deyvani | 0.541 ± 0.051 c | 68.86 ± 3.60 a | 5.14 ± 0.44 a | 0.52 ± 0.07 b |
| | Haseni | 0.586 ± 0.066 b | 68.59 ± 6.22 a | 4.53 ± 0.33 b | 0.67 ± 0.04 a |
| | Reşek | 0.662 ± 0.096 a | 66.91 ± 4.56 a | 4.99 ± 0.27 a | 0.55 ± 0.05 b |
| **Drying method** | | | | | |
| | AOW | 0.618 ± 1.026 a | 70.86 ± 4.26 a | 4.91 ± 0.56 a | 0.56 ± 0.09 b |
| | POT | 0.569 ± 0.076 b | 68.63 ± 4.28 ab | 4.97 ± 0.46 a | 0.57 ± 0.11 b |
| | SD | 0.602 ± 0.037 ab | 64.88 ± 4.24 b | 4.78 ± 0.26 a | 0.62 ± 0.04 a |
| **Variety × drying method** | | | | | |
| | AOW | 0.563 ± 0.032 de | 68.01 ± 3.90 abc | 5.34 ± 0.31 ab | 0.51 ± 0.03 de |
| Deyvani | POT | 0.483 ± 0.028 f | 70.35 ± 4.04 ab | 5.43 ± 0.31 a | 0.46 ± 0.03 e |
| | SD | 0.577 ± 0.033 cd | 68.24 ± 3.92 abc | 4.66 ± 0.27 cde | 0.60 ± 0.03 c |
| | AOW | 0.512 ± 0.029 ef | 73.52 ± 4.22 a | 4.25 ± 0.24 e | 0.67 ± 0.04 ab |
| Haseni | POT | 0.646 ± 0.037 b | 70.35 ± 4.04 ab | 4.52 ± 0.26 de | 0.71 ± 0.04 a |
| | SD | 0.601 ± 0.034 bcd | 61.91 ± 3.55 c | 4.81 ± 0.28 cd | 0.64 ± 0.04 bc |
| | AOW | 0.779 ± 0.045 a | 71.05 ± 4.08 ab | 5.14 ± 0.29 abc | 0.51 ± 0.03 de |
| Reşek | POT | 0.579 ± 0.033 cd | 65.19 ± 3.74 bc | 4.95 ± 0.28 abcd | 0.54 ± 0.03 d |
| | SD | 0.629 ± 0.036 bc | 64.49 ± 3.70 bc | 4.88 ± 0.28 bcd | 0.60 ± 0.03 c |
| **ANOVA** | | | | | |
| $F_{Variety}$ | | 28.18 *** | 0.66 ns | 11.82 *** | 50.17 *** |
| $F_{Drying\ method}$ | | 4.64 * | 5.37 * | 1.00 ns | 6.74 ** |
| $F_{Variety\ \times\ drying\ method}$ | | 20.31 *** | 2.28 ns | 4.65 ** | 8.46 *** |

Different letters in the same column indicate a significant difference between means according to Student's *t*-test ($p < 0.05$) for each factor. ns: non-significant. *, **, and *** indicate significance at $p < 0.05$, $p < 0.01$, and $p < 0.001$, respectively.

The total soluble solids (TSS) showed relatively low variation across the different varieties, whereas there was a significant variation among the drying methods ($p < 0.05$). The interaction between variety and drying method was found to be insignificant. The pH levels significantly differed among the varieties ($p < 0.001$), with Haseni having the lowest pH of 4.53. The interaction between variety and drying method was also significant ($p < 0.01$), and in the POT treatment, the Deyvani variety exhibited the highest pH at 5.43.

However, the pH values of the drying methods ranged from 4.78 to 4.91, and this variation was not significant. In contrast to the pH, the titratable acidity (TA) showed the opposite trend, which is an expected occurrence, except for the significance of the drying method. The highest TA value of 0.71 mg/L was observed in the Haseni variety with the POT treatment, while the lowest value was found in the Deyvani variety, which also underwent the POT treatment (Table 1).

### 3.2. Total Phenolics and DPPH Scavenging

The total phenolic content was significantly influenced by the variety, drying method, and their interactions ($p < 0.001$). The total phenolics ranged from 82.29 mg/L GAE (Reşek) to 104.30 mg/L GAE (Deyvani) across the different varieties. Among the drying methods, the POT treatment exhibited notably higher total phenolic content (118.37 mg/L GAE) compared to the SD treatment (78.24 mg/L GAE) and the AOW treatment (80.43 mg/L GAE). The highest mean interaction value was observed in the POT treatment for the Deyvani variety. It is worth mentioning that the phenolic content was consistently most elevated in the POT treatment for all varieties, while the lowest amounts varied (Figure 1).

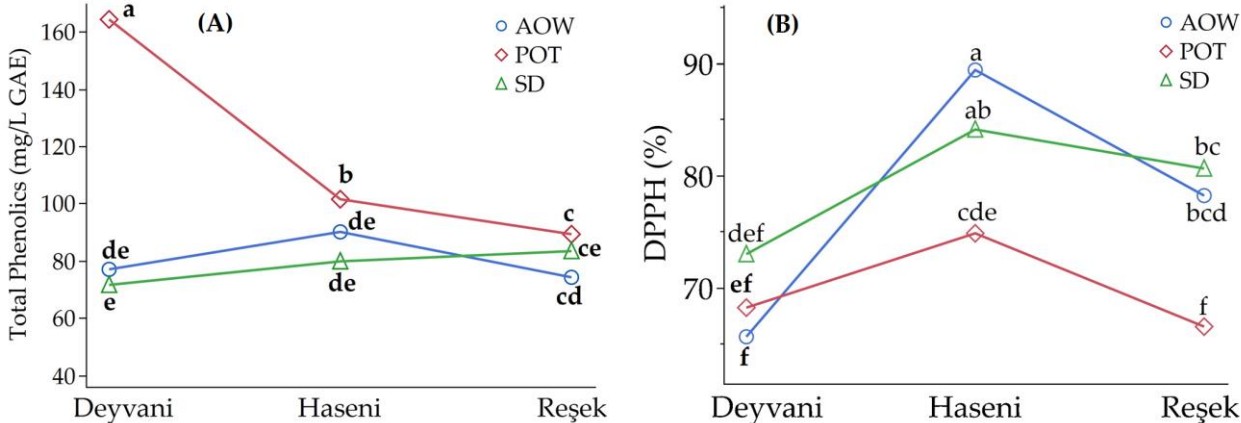

**Figure 1.** Fluctuations in the total phenolics content (**A**) and DPPH scavenging activity (**B**) according to the varieties and drying methods. The letters above the figures indicate significant differences in Student's *t*-test ($p < 0.05$). AOW: ashy-oily water; POT: potassium carbonate solution; SD: sun drying.

The DPPH scavenging capacity displayed significant variations among the varieties ($p < 0.001$), drying methods ($p < 0.001$), and their interactions ($p < 0.05$). Among the varieties, the Haseni variety exhibited the highest DPPH scavenging activity (83%), whereas the Deyvani variety had the lowest activity, with a scavenging capacity of 69%. The differences in DPPH scavenging capacity across the treatments were relatively modest compared to the variations among the varieties. The POT treatment showed the lowest DPPH scavenging activity at 70%, while the AOW treatment had 78% and the SD treatment had an average of 79%. The interactive effect on the DPPH scavenging capacity yielded diverse outcomes. The Haseni variety treated with AOW demonstrated the highest activity at 89%, while the AOW-treated Deyvani and POT-treated Reşek exhibited the lowest activity with an average of 66% (Figure 1).

### 3.3. Changes in Protein and Anthocyanin Content

The soluble protein content showed significant differences among the varieties, drying methods, and their interactions ($p < 0.001$). The soluble protein content ranged from 1.07 mg/mL (Haseni) to 2.49 mg/mL (Reşek), indicating at least a 2-fold difference in protein content across the different varieties. In terms of the drying methods, the AOW and SD treatments were similar in protein content, with 1.18 mg/mL and 1.13 mg/mL, respectively. However, the POT treatment displayed over 2-fold higher protein content, with an average of 2.47 mg/mL. When evaluating the interactive effect, the Deyvani and

Reşek varieties exhibited the highest protein content in the POT treatment, with mean values of 2.08 mg/mL and 4.25 mg/mL, respectively, while the Haseni variety had the highest protein content in the SD treatment, with a mean of 1.31 mg/mL (Figure 2, Table S1).

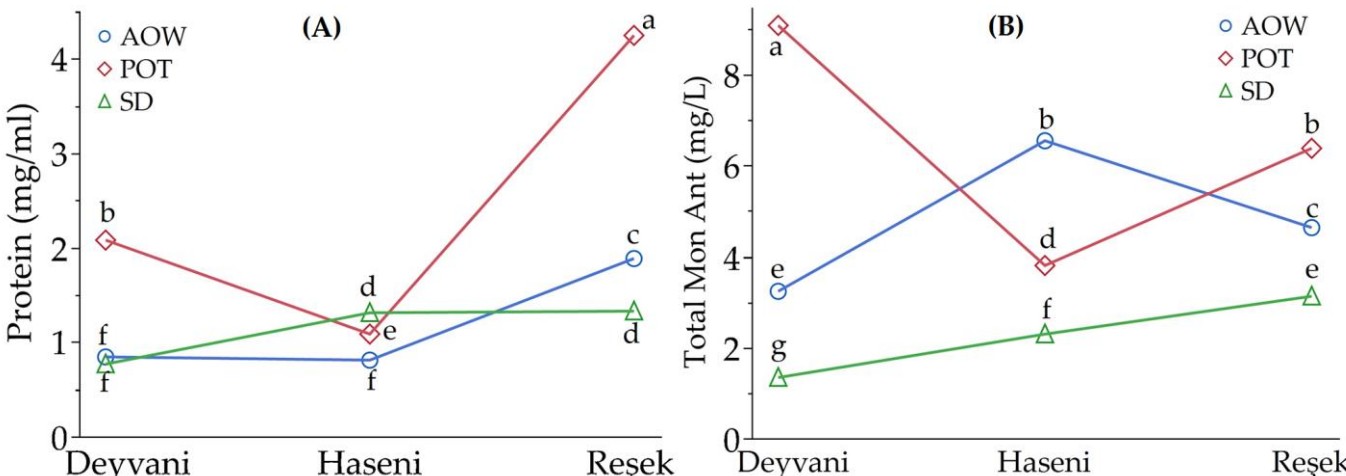

**Figure 2.** Changes in protein (**A**) and total monomeric anthocyanin (**B**) content according to the variety and drying method. The letters above the figures indicate significant differences in Student's *t*-test ($p < 0.05$). AOW: ashy-oily water; POT: potassium carbonate solution; SD: sun drying.

The total monomeric anthocyanin content exhibited significant variations based on the variety ($p < 0.01$), drying method ($p < 0.001$), and their interaction ($p < 0.001$). The differences in mean monomeric anthocyanin values among the varieties were relatively narrow, ranging from 4.22 mg/L (Haseni) to 4.72 mg/L (Reşek). However, sun-dried raisins had two to three times lower monomeric anthocyanin content than AOW and POT-treated raisins. AOW-treated raisins contained 4.81 mg/L, POT-treated raisins had 6.42 mg/L, and sun-dried raisins had only 2.25 mg/L of monomeric anthocyanin. The interactions revealed a completely different pattern. The treatments led to a significant diversity of monomeric anthocyanin in the Deyvani variety, ranging from 1.34 mg/L (SD) to 9.08 mg/L (POT). Reşek also exhibited a similar trend within a relatively narrow range. On the other hand, the Haseni variety displayed the highest monomeric anthocyanin content in the AOW treatment (6.55 mg/L), while the lowest was in the SD treatment (2.29 mg/L), as observed in the other varieties (Figure 2, Table S1).

*3.4. Changes in Raisin Color Properties*

The color parameters, including L, a, b, Chroma, and Hue, were significantly influenced by the variety, drying method, and their interactions. However, the degree of influence varied depending on the specific color attribute. The interaction effect was the least significant (based on the F value) for all color properties except for Hue, while the drying method had the greatest impact on all color properties. Notably, the effect of drying methods on the L value and Hue angle was found to be at least twice as significant as the other two factors. Furthermore, when analyzing the changes in color values based on drying methods for different varieties, it is evident that white varieties exhibit a wider range of color changes due to drying (Table 2).

In terms of color intensity, the Reşek variety displayed the most intense color, followed by Deyvani and Haseni, in that order. There was no statistically significant difference in average color intensity between the SD and POT drying methods, while the AOW treatment yielded lower color density compared to the other two methods. The Haseni and Reşek varieties achieved the highest color intensity with the SD treatment, while Deyvani showed the highest intensity with the POT treatment. Moreover, the Reşek variety exhibited the highest values in terms of polymeric color among the different drying methods, with the

POT treatment yielding the highest values overall. All varieties displayed the highest polymeric color when subjected to the POT treatment (Table 2).

**Table 2.** Changes in color features of raisins according to the variety, drying method, and their interaction (mean ± standard deviation).

| Variety | | L* | a* | b* | Chroma | Hue$^{o}$ | ColDen | PolCol |
|---|---|---|---|---|---|---|---|---|
| | Deyvani | 19.79 ± 1.98 a | 3.75 ± 1.41 a | 4.75 ± 2.52 a | 6.09 ± 2.81 a | 48.72 ± 7.39 a | 2.12 ± 0.16 b | 1.30 ± 0.23 c |
| | Haseni | 17.74 ± 1.57 b | 3.15 ± 1.36 b | 3.82 ± 1.70 b | 4.96 ± 2.18 b | 50.14 ± 2.76 a | 1.75 ± 0.16 c | 1.57 ± 0.09 b |
| | Reşek | 19.26 ± 1.84 a | 1.33 ± 0.71 c | 1.22 ± 0.99 c | 2.00 ± 0.94 c | 42.59 ± 11.11 b | 2.70 ± 0.32 a | 2.17 ± 0.48 a |
| Drying method | | | | | | | | |
| | AOW | 18.04 ± 0.98 b | 3.20 ± 1.21 b | 3.83 ± 1.23 b | 5.00 ± 1.72 b | 51.92 ± 4.47 a | 2.04 ± 0.30 b | 1.46 ± 0.37 c |
| | POT | 18.23 ± 2.02 b | 3.79 ± 1.42 a | 4.81 ± 2.71 a | 6.18 ± 2.96 a | 48.92 ± 8.47 b | 2.25 ± 0.54 a | 1.97 ± 0.63 a |
| | SD | 20.52 ± 1.72 a | 1.25 ± 0.68 c | 1.15 ± 0.87 c | 1.87 ± 0.87 c | 40.61 ± 7.29 c | 2.29 ± 0.50 a | 1.60 ± 0.24 b |
| Variety × drying method | | | | | | | | |
| Deyvani | AOW | 17.65 ± 1.01 cd | 4.30 ± 0.25 b | 4.87 ± 0.28 c | 6.51 ± 0.37 b | 48.68 ± 2.79 b | 2.11 ± 0.12 bc | 1.02 ± 0.06 e |
| | POT | 20.15 ± 1.16 ab | 5.00 ± 0.29 a | 7.58 ± 0.43 a | 9.09 ± 0.52 a | 56.80 ± 3.26 a | 2.26 ± 0.13 b | 1.49 ± 0.09 cd |
| | SD | 21.57 ± 1.24 a | 1.93 ± 0.11 d | 1.80 ± 0.10 f | 2.68 ± 0.15 de | 40.70 ± 2.34 c | 2.00 ± 0.11 c | 1.39 ± 0.08 d |
| Haseni | AOW | 18.54 ± 1.06 bc | 3.64 ± 0.21 c | 4.39 ± 0.25 d | 5.71 ± 0.33 c | 50.41 ± 2.89 b | 1.69 ± 0.10 d | 1.53 ± 0.09 cd |
| | POT | 16.01 ± 0.92 d | 4.41 ± 0.25 b | 5.42 ± 0.31 b | 7.00 ± 0.40 b | 51.38 ± 2.95 b | 1.64 ± 0.09 d | 1.63 ± 0.09 c |
| | SD | 18.68 ± 1.07 bc | 1.41 ± 0.08 e | 1.66 ± 0.10 f | 2.18 ± 0.13 e | 48.65 ± 2.79 b | 1.92 ± 0.11 c | 1.54 ± 0.09 cd |
| Reşek | AOW | 17.92 ± 1.03 c | 1.65 ± 0.09 de | 2.23 ± 0.13 e | 2.78 ± 0.16 d | 56.69 ± 3.25 a | 2.32 ± 0.13 b | 1.84 ± 0.11 b |
| | POT | 18.54 ± 1.06 bc | 1.95 ± 0.11 d | 1.44 ± 0.08 f | 2.45 ± 0.14 de | 38.58 ± 2.21 c | 2.85 ± 0.16 a | 2.79 ± 0.16 a |
| | SD | 21.31 ± 1.22 a | 0.40 ± 0.02 f | 0.10 ± 0.01 g | 0.76 ± 0.04 f | 32.50 ± 1.86 d | 2.94 ± 0.17 a | 1.89 ± 0.11 b |
| ANOVA | | | | | | | | |
| F$_{Variety}$ | | 8.52 ** | 439.40 *** | 575.49 *** | 472.80 *** | 19.30 *** | 126.04 *** | 180.41 *** |
| F$_{Drying\ method}$ | | 14.42 *** | 490.21 *** | 618.58 *** | 523.00 *** | 41.06 *** | 9.86 ** | 62.58 *** |
| F$_{Variety\ \times\ drying\ method}$ | | 4.69 ** | 18.26 *** | 75.99 *** | 51.99 *** | 24.36 *** | 8.99 *** | 22.18 *** |

Different letters in the same column indicate a significant difference between means according to Student's $t$-test ($p < 0.05$) for each factor. ** and *** indicate significance at $p < 0.01$ and $p < 0.001$, respectively.

### 3.5. Statistical Approaches

In the PCA analysis, the first two components accounted for 59.8% of the total variability, with PC1 contributing 36.7% and PC2 contributing 23.1%. The variables that had the highest influence on PC1 were a*, b*, and Chroma, with vector loadings of 0.408, 0.424, and 0.424, respectively. On the other hand, PC2 was primarily influenced by protein content (0.458) and color density (0.430) (Table S2). The PC plot revealed distinct regions for each variety, indicating their separation based on the analyzed variables. Similarly, the drying methods were also segregated into different zones on the plot. In terms of the relationships

observed, the POT treatment possessed high contents of total monomeric anthocyanin, total phenolics, and pH. The SD treatment, on the other hand, was aligned with the berry weight. AOW treatment exhibited relatively stronger connections with TSS and Hue angle (Figure 3). These findings highlight the distinct contributions of different variables and their impact on the separation and relationships observed in the PCA analysis.

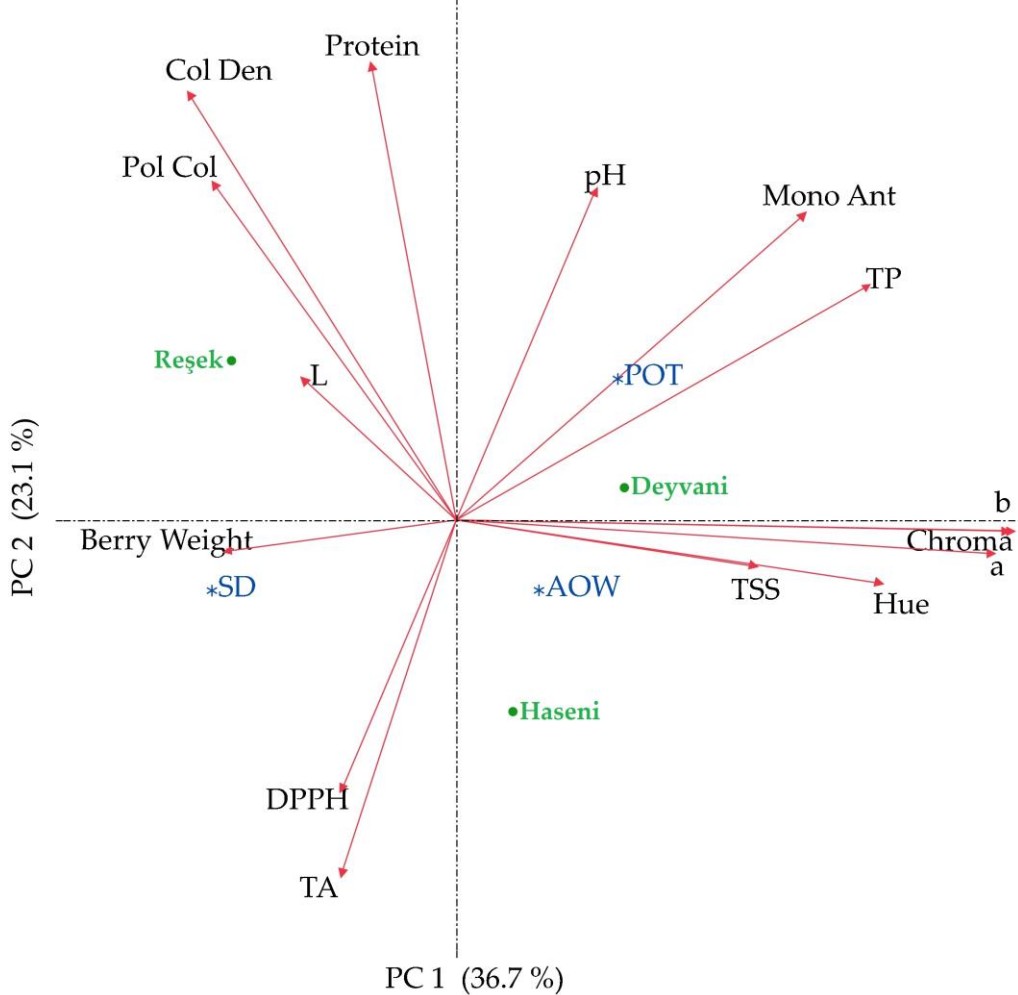

**Figure 3.** Biplot PCA analysis illustrating the distribution of the varieties and drying methods according to the relationships across the studied traits. AOW: ashy-oily water; POT: potassium carbonate solution; SD: sun drying. Col Den: color density; Pol Col: polymeric color; TA: titration acidity; Mono Ant: total monomeric anthocyanin; TP: total phenolic content; TSS: total soluble solids. The * represents treatments while the • represents varieties.

The heatmap analysis visually demonstrated the variations in different traits across the varieties and drying methods. Specifically, it revealed that the Deyvani variety treated with POT exhibited the highest values in a*, b*, Hue, Chroma, pH, total phenolics, and total monomeric anthocyanin content. In contrast, the SD treatment of the Deyvani variety resulted in lower values for these same features. A similar pattern was observed in the Haseni variety, where the SD treatment showed lower values compared to the AOW and POT treatments. On the other hand, the changes in traits for the Reşek variety differed from those of the other two varieties across the different treatments (Figure 4).

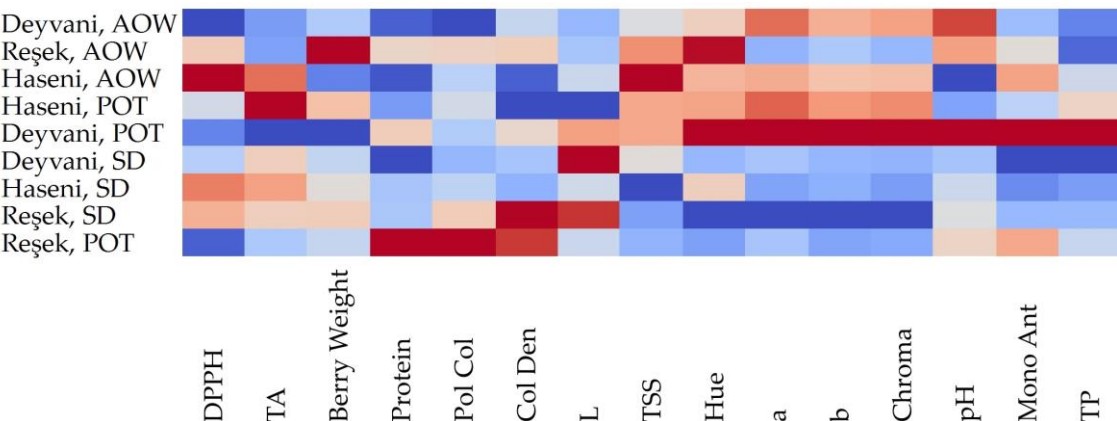

**Figure 4.** The heatmap illustrates the changes in the studied characteristics in terms of the varieties and drying methods. The color scale from blue to red indicates low to high values for each trait. AOW: ashy-oily water; POT: potassium carbonate solution; SD: sun drying. Col Den: color density; Pol Col: polymeric color; TA: titration acidity; Mono Ant: total monomeric anthocyanin; TP: total phenolic content; TSS: total soluble solids.

## 4. Discussion

Grape skin is composed of an epidermis and several layers of small thick-walled cells, which play a crucial role in regulating the drying process. The outer epidermis is covered by non-living layers, including cuticles, lenticels, wax, and collenchymatous hypodermal cells [37]. The wax layer, due to its hydrophobic nature, forms a protective barrier against fungal pathogens and minimizes water loss through transpiration [38]. Additionally, it shields the grape from UV light and physical damage [39]. The skin also controls the exchange of gases between the berry and its surrounding environment. The epicuticular layer, known for its role in protecting against microbial and environmental damage, can influence the shelf life of all grape-related products [23]. The number, size, and volume of these skin layers vary depending on the grape cultivar [40]. The variation of the grape skin characteristics, particularly cuticular wax abundance and composition, could be one of the main reasons for the significantly different berry weights and physicochemicals across the drying methods.

A recent study reported no significant change in TSS and pH due to drying methods for the Bilbizeki, Raşe Kewnar, and Kerküş varieties, while there was a significant difference in the berry weight of the Kerküş variety according to the drying treatment [41]. According to Jadhav et al. [42], the 'Thompson seedless' grape variety dipped in a solution of potassium carbonate and olive oil exhibited a soluble solid content (SSC) ranging from 70.71% to 82.97% and titratable acidity ranging from 0.25% to 0.32%. The acidity levels of raisins obtained through various dipping solutions and drying techniques ranged from 0.46% to 0.51% in the 'Thompson seedless' variety and from 0.54% to 0.63% in the 'Perlette' variety [43]. Yalcinkaya [44] observed pH values ranging from 4.29 to 4.52, acidity values between 0.77% and 1.05%, and TSS between 81.74% and 87.13% in raisins subjected to different drying methods, which were lower for pH but higher for TA and TSS. The variation in acidity could be due to the drying technique, growing ecology, and variety [45].

The antioxidant properties of foods are associated with their ability to inhibit or scavenge reactive oxygen species (ROS). The DPPH assay has gained popularity due to its easy, rapid, and cost-effective application [46]. This technique relies on the elimination of DPPH free radicals by acting as a hydrogen donor to ROS [47]. It is important to note that this method is sensitive to acidic pH levels. Higher hydrogen ion concentrations can lead to a decrease in the antioxidant activity of the sample [48]. In this study, the Deyvani and Reşek varieties exhibited similar pH levels across the treatments, resulting in relative changes in DPPH scavenging activity. However, the Haseni variety consistently displayed the highest DPPH activity across all treatments, indicating that the antioxidant properties of

raisins are influenced by the specific grape variety, emphasizing the importance of selecting an appropriate variety. The findings of this study are in line with the findings of Breska et al. [27], who reported variations in antioxidant levels among various grape cultivars in raisins. Additionally, Keskin et al. [4] observed higher antioxidant enzyme activities in raisins treated with oak ash compared to those treated with potassium carbonate in the Gök Üzüm variety. In the present study, AOW treatment demonstrated higher antioxidant activities than POT treatment in the Haseni and Reşek varieties, while the Deyvani variety exhibited higher activity in POT treatment. These results suggest that the change in antioxidant properties in raisins is a complex process influenced by various factors.

The majority of macromolecules in raisins are carbohydrates, followed by protein and fat. In Tunisian grape raisins, the protein content was reported to vary among varieties, ranging from 1.65% to 3.33% [49]. Conversely, Zemni et al. [50] reported protein contents not exceeding 1.3% and found no significant difference in protein content based on the drying method in the Italian Muscat cultivar. Previous researchers used the Kjeldahl assay to quantify protein content, while in this study, the more sensitive Bradford assay was employed [51]. The conflicting statements among previous researchers and the findings of this study could be due to differences in cultivars, measurement assays, or pretreatment conditions.

Color is a crucial aspect of both processed and non-processed foods, playing a vital role in consumer acceptance of the product. It can also indicate chemical changes that occur during production and storage. Researchers have highlighted the importance of color changes in dried grapes as a quality indicator, influenced by various chemical and biochemical processes during drying and storage [52]. Compared to other food items, raisins are more prone to color changes, especially in terms of browning reactions, whether enzymatic or non-enzymatic, as well as due to storage and distribution conditions [53]. Non-enzymatic browning reactions involve the interaction between amino acids or proteins and sugars, resulting in the formation of nitrogen polymers and a darkening of the product. Consumers generally prefer grapes or raisins with brighter colors, making it important to prevent browning reactions that lead to darkening [54] and undesirable flavors in raisins. In this study, the L* values were comparable to those in previous studies on raisins [55]. On the other hand, the a* and b* values of all raisin varieties were positive, indicating the presence of colors ranging from red to yellow. As expected, the black Reşek variety exhibited a higher a* value than b*, while the green varieties had higher b* values. Interestingly, sun-dried raisins showed the highest brightness across all varieties, suggesting a lower degree of oxidation. Browning in raisins is typically caused by enzymatic reactions and the breakdown of cellular structures [56]. The results of this study, particularly the limited variation in a* and b* values, highlight the importance of bleaching treatments to achieve specific colors for each variety of raisins.

The PCA and heatmap analysis have started to be utilized in research studies to evaluate the differentiation of the factors according to the studied characteristics from the visual point of view [57,58]. By examining the biplot, it is possible to observe the relative positions of the varieties and drying methods, indicating their similarities or differences in terms of the studied traits. The biplot can also reveal which traits have the most significant influence on the separation of varieties and drying methods. Overall, the biplot serves as a visual tool to aid in understanding the complex relationships and patterns among the variables and how they contribute to the distribution of varieties and drying methods in the analyzed dataset. The heatmap analysis provided a visual representation of these variations, allowing for easy comparison and identification of the differences between varieties and drying methods. The heatmap analysis revealed distinct differences in trait changes for the Reşek variety compared to the other two varieties across different treatments. Moreover, the pretreatments were grouped together, except for the POT-treated Reşek variety, which indicates that treatments significantly affected distinctions among raisins. Additionally, the biplot analysis shows a clear separation among all treatments and varieties, indicating proximate variation in raisin characteristics of the varieties according to the treatments.

### 5. Conclusions

This study aimed to assess the impact of different drying methods on the changes observed in raisins from specific grape varieties. The drying methods included no pretreatment, pretreatment with potassium carbonate, and pretreatment with ashy-oily water. The results revealed significant variations in phenolic content, antioxidant capacity, color properties, berry weight, and physicochemical characteristics, which were influenced by both the variety of grapes and the drying methods. The Deyvani variety exhibited the highest phenolic content, antioxidant capacity, and color properties when treated with potassium carbonate, while the Reşek variety had higher levels of protein, color density, and polymeric color. In contrast, the Haseni variety showed the highest antioxidant activity and color properties when treated with ashy-oily water. The color density was predominantly influenced by the grape variety, highlighting the importance of selecting the appropriate variety for market purposes. Future research should focus on evaluating quality changes and bioactive compounds during shelf life while considering the interaction between variety and pretreatment.

**Supplementary Materials:** The following supporting information can be downloaded at https://www.mdpi.com/article/10.3390/horticulturae9070771/s1: Table S1: The alteration of total phenolics, DPPH, total monomeric anthocyanin and protein content according to the varieties, drying methods, and their interaction (mean ± standard deviation), Table S2. Eigenvectors for the studied features.

**Author Contributions:** Conceptualization, M.S.Ü. and E.G.; methodology, M.S.Ü., E.G. and M.Y.; software, E.G.; formal analysis, E.G. and M.Y.; investigation, E.G., M.S.Ü. and M.Y.; resources, M.S.Ü.; data curation, E.G.; writing—original draft preparation, E.G., M.S.Ü. and M.Y.; writing—review and editing, E.G., M.S.Ü. and M.Y.; visualization, E.G. All authors have read and agreed to the published version of the manuscript.

**Funding:** This research received no external funding.

**Data Availability Statement:** The data for this study are available from the corresponding author upon reasonable request.

**Conflicts of Interest:** The authors declare no conflict of interest.

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
