# Peer review of "Changes in Antioxidant and Color Properties of Raisins According to Variety and Drying Method"

_horticulturae, doi:10.3390/horticulturae9070771_

Round 1

Reviewer 1 Report

In this paper, the effects of different pretreatment on physicochemical, antioxidant and colorimetric properties of different dried grape varieties in high altitude area of Turkey were studied, so as to provide reference for the development of dried grape varieties suitable for current ecology. It is worth noting that in this paper, only samples under the two pretreatment methods and untreated control samples were compared for drying, which is a comparison of pretreatment methods, including natural drying, hot air drying, vacuum freeze-drying, etc., while the drying methods in this paper were all natural drying, so there was no comparison of different drying methods. It is suggested to modify and improve the title, content and chart of the article; In addition, the overall analysis of this paper is not clear enough, only stays on the surface of the data, does not explore the deep causes of changes, lacks logic and coherence, and some languages have ambiguity, so it is suggested to modify and improve. The figures format of the full text is not uniform. To sum up and based on the following reasons, it is suggested to modify and improve the current situation.

The main comments and notes are in the appendix below:

Comments:

1. Line 176-184

The interactions mentioned here are not detailed, and the reasons for the changes in the weight and pH of raisins and the advantages and disadvantages of each variety are not shown.

2. Fig. 1 and Fig. 2

It is recommended to label the two diagrams separately, and explain the name of each diagram separately, denoted by (A) and (B). In addition, the differences between the three varieties are compared here, and it is not recommended to make a line chart to illustrate.

3. Lines 244-245

What's meaning of the interactions revealed a completely different pattern?

4. Fig. 3

The sum of the two principal components in the PCA diagram does not reach 60%, can it reflect the overall information of the sample.

5. Lines 290-292

The expression that POT treatment is linearly correlated with total monomer anthocyanin content, total phenols and pH is incorrect, which could only indicate that the total monomer anthocyanin content and total phenolic substances in raisins after POT treatment were more and related to pH.

6. Lines 391-405

The introduction of PCA and heat maps should be used to fully explain the differences between different pretreatments and different grape varieties, and should not prove the advantages of the two data analysis methods at the conclusion.

 Minor editing of English language required

Reviewer 2 Report

Dear authors, after reviewing the submitted article "The alteration of raisin characteristics of certain varieties according to drying methods", I would like to make the following recommendations to improve your work:

Title is very general, what are those characteristics important for raisins, it needs a more specific title.

Line 11, the methods are implemented in the production process have any relevance or what is the relevant point of the sentence?

Lines 12 and 13, the abbreviations of the treatments should be clear to identify what each treatment consisted of.

Lines 23 to 25 emphasize that variety is crucial so the title should focus on that.

In key words consider the most important ones for the tracking of the article.

The introduction talks about the spill and the economic value of the product in the world in addition to the nutritional importance, I recommend including data on the nutritional composition of raisins, more importance is given to the antioxidant part, but it is not the only thing that was evaluated.

On the other hand, it does not delve into the methods currently used or the traditional methods historically used in the raisin production region.

On line 59 is Thompson

Lines 59 to 64 do not understand what idea the authors want to transmit, it seems that words are missing.

Line 63, in what context is the term ecology used? I consider that it is not adequate for what you want to explain.

Line 66 is vital to what?

In the section of materials and methods, it was not considered to compare the results of the Sultani Çekirdeksiz variety with the varieties studied?

Line 82, indicates v/v is that correct? does it not correspond to w/v?

Why were those solutions selected? it would be convenient to put a background

Line 99, did the trials have replicates or repetitions?

Line 112, the sentence is not understood.

Line 130, why is the protein value important for the assay?

Line 142, +4 it is correct?

Line 166, triplicates for each treatment means that you used 3 batches to analyze in each one the raisins obtained?

Line 180 has an error, it is not 6.18 g.

In Figure 4, it is not necessary to indicate the color intensity scale?

In the discussion section, you need to deepen the explanation of the results they obtained, increase the citations of contrasting information to validate their results.

Lines 330 to 341, corresponds to material for the introduction section.

The authors should explain the effect of the components of the solutions in their results, only mentions are made but it is not explained at the molecular level.

The article in general lacks an explanation of the reason for their results beyond repeating that the grape variety plays a crucial role when transformed into raisins.

I suggest the authors to revise the article again and rely on significant references to explain the results.

Best regards.

Author Response

As authors, we appreciate the Reviewer for his/her valuable suggestions to improve the quality of the paper. After employing the corrections, he/she suggested, we clearly see the article was comprehensively improved. Our responses to the reviewer’s queries are below;

Q1: Title is very general, what are those characteristics important for raisins, it needs a more specific title.

A1: The title of the manuscript was changed as “Changes in antioxidant and color properties of raisins according to varieties and drying methods”.

Q2: Line 11, the methods are implemented in the production process have any relevance or what is the relevant point of the sentence?

A2: The sentence was used as an introduction to drying methods but lacks the meaning. Therefore, it was changed to “There are various methods employed for drying grapes into raisins.”

Q3: Lines 12 and 13, the abbreviations of the treatments should be clear to identify what each treatment consisted of.

A3: The treatment “ashy water including %2.5 olive oil (AOW)” had been miswritten. It was corrected. The other treatments’ abbreviations are relevant to their names. Thanks a lot for the suggestion.

Q4:  Lines 23 to 25 emphasize that variety is crucial so the title should focus on that.

A4: The variety factor was added to the title.

Q5: In key words consider the most important ones for the tracking of the article.

A5: The keywords were chosen from the essential words that title does not include to provide a wide range of indexing.

Q6: The introduction talks about the spill and the economic value of the product in the world in addition to the nutritional importance, I recommend including data on the nutritional composition of raisins, more importance is given to the antioxidant part, but it is not the only thing that was evaluated. On the other hand, it does not delve into the methods currently used or the traditional methods historically used in the raisin production region.

A6: A new paragraph was added about the methods of raisin production to the Introduction in line with your suggestions. The nutritional value of raisins was also added to the Introduction.

Q7: On line 59 is Thompson

A7: Corrected

Q8: Lines 59 to 64 do not understand what idea the authors want to transmit, it seems that words are missing.

A8: That part of the introduction states the necessity of the determination of new varieties for raisin production in various ecologies.

Q9: Line 63, in what context is the term ecology used? I consider that it is not adequate for what you want to explain.

A9: Thanks for your attention. The ecology term had been misused. We tried to indicate the geo-climatic conditions and the necessity of the use of proper variety. The term was corrected as geo-climate.

Q10: Line 66 is vital to what?

A10: Corrected as “vital to obtain desired quality raisins.”

Q11: In the section of materials and methods, it was not considered to compare the results of the Sultani Çekirdeksiz variety with the varieties studied?

A11: Sultani Çekirdeksiz is not grown in the area that studied grape varieties are cultivated. Thus, we did not include Sultani Çekirdeksiz to the study to eliminate the possible effects of the geographic and climatic conditions.

Q12: Line 82, indicates v/v is that correct? does it not correspond to w/v?

A12: It should be w/v, and was corrected. Thanks a lot.

Q13: Why were those solutions selected? it would be convenient to put a background

A13: These solutions are widely used in Turkey. However, as you mentioned, we did not indicate their importance in the manuscript. The relevant sentence about why we chose these treatments was added to introduction, in the part explaining the importance of the drying methods that we newly added.

Q14: Line 99, did the trials have replicates or repetitions?

A14: Trials were carried out triplicates. We stated it in the statistics section.

Q15: Line 112, the sentence is not understood.

A15: The sentence was rewrited as “The microscale procedure reported by Waterhouse [23] was used with some modifications in the determination of the total phenolic content”

Q16: Line 130, why is the protein value important for the assay?

A16: As a healthy snack, raisins contain good amount of proteins (approximately 3%). Thus, we thought its alteration deserves an attention.

Q17: Line 142, +4 it is correct?

A17: The method says store in the refrigerator for 2 hours. It is correct but converted to “kept in refrigerator  for 2 hours in dark conditions”

Q18: Line 166, triplicates for each treatment means that you used 3 batches to analyze in each one the raisins obtained?

A18: It means all factors had 3 different batches when processing grapes into raisins.

Q19: Line 180 has an error, it is not 6.18 g.

A19: It is 0.618 g not 6.18 g. It had been miswritten. Thanks for your attention.

Q20: In Figure 4, it is not necessary to indicate the color intensity scale?

A20: The color legend values are different for each trait. On the other hand, you are right about the necessity of the indication of color scale.  The sentence about the color scale was added as “The color scale from blue to red indicates low to high values for each trait.” Thanks for your attention.

Q21: In the discussion section, you need to deepen the explanation of the results they obtained, increase the citations of contrasting information to validate their results.

A21: The discussion part was strengthened according to the suggestions.

Q22: Lines 330 to 341, corresponds to material for the introduction section.

A22: Thanks for the notification. The relevant part was merged with the introduction and excluded from the discussion.

Q23: The authors should explain the effect of the components of the solutions in their results, only mentions are made but it is not explained at the molecular level.

A23: The discussion part was strengthened in line with your suggestions

Q24: The article in general lacks an explanation of the reason for their results beyond repeating that the grape variety plays a crucial role when transformed into raisins.

A24: The article was improved in line with your suggestions by discussing results in different ways.

Reviewer 3 Report

Raisin production is an important sector of viticulture therefore research studies dealing with the production and methodologies are important and very welcome.

The introduction contains information about the history and present state of the sector. In my opinion, the order of the content should be changed, and first the history later the present state should be interpreted.

I miss the “gap” from the introduction as no information about the existed drying methodology is included. It would be important to include why the experiments are necessary and what is the question to be answered.

Materials and Methods:

Please specify the “adequate number of grapes” in kg and/or number of bunches. Please explain how the solutions were applied and how the bunches/berries were treated with the solutions. How many replications were applied? For how long the berries were dried?

Weight of the berries are introduced, but no information about the berry size is given. Was there any difference among the cultivars in berry size? There are noticeably differences in size among berries originating from different part of the bunches (bottom, middle, top). From which position of the bunched were the berries collected?

Please give a detailed citation of the colorimeter as there are many models provided by the PCE.

Discussion: Please give examples for studies where different drying methods are examined. Discussion is detailed, but only in the context of the investigated parameters and those differences among cultivars. It would be important to discuss the drying methods too.

With the above mentioned minor corrections, I suggest the study to be published. 

The quality of the study is appropriate for publication. 

Author Response

We appreciate the reviewer for his/her precious comments to improve the quality of the paper. Our responses to reviewer’s queries are below:

Q1: The introduction contains information about the history and present state of the sector. In my opinion, the order of the content should be changed, and first the history later the present state should be interpreted.

A1: Introduction was reorganized in line with the suggestions of both reviewers

Q2: I miss the “gap” from the introduction as no information about the existed drying methodology is included. It would be important to include why the experiments are necessary and what is the question to be answered.

A2: The information about the methods were added to Introduction.

Q3: Please specify the “adequate number of grapes” in kg and/or number of bunches. Please explain how the solutions were applied and how the bunches/berries were treated with the solutions. How many replications were applied? For how long the berries were dried?

A3: The preparation of the solutions and application was explained in the materials and methods section.

Q4: Weight of the berries are introduced, but no information about the berry size is given. Was there any difference among the cultivars in berry size? There are noticeably differences in size among berries originating from different part of the bunches (bottom, middle, top). From which position of the bunched were the berries collected?

A4: The varieties selected for the study had almost same weighted berries. The bunches that had uniform berry sizes were selected for the study. It was included in the materials and method section.

Q5: Please give a detailed citation of the colorimeter as there are many models provided by the PCE.

A5: Given as “PCE CSM-4, UK”.

Q6: Discussion: Please give examples for studies where different drying methods are examined. Discussion is detailed, but only in the context of the investigated parameters and those differences among cultivars. It would be important to discuss the drying methods too.

A6: Discussion part was strengthened by the previous studies.

Best regards

Round 2

Reviewer 2 Report

Dear authors, after reviewing again your article I consider that it meets the requirements and the quality necessary for its publication. I appreciate that every suggestion has been incorporated into the article to improve its presentation, comprehension and impact on knowledge.

Best regards.